# Molecular signature evolution of coal-derived dissolved organic matter under geothermal conditions: FT-ICR MS and machine learning

Peng Ge[1], Tan Liu[1], Dong Dong[1], Zepeng Wan [2]*, Yanqing Wu[2]*

1 China Coal Technology & Engineering Group Hangzhou Research Institute Co., Ltd., Hangzhou, China,
2 School of Resources and Safety Engineering, Chongqing University, Chongqing City, China

* wanzepeng1996@163.com (ZW); wuyanqing9@163.com (YW)

## Abstract

The accurate identification of water sources and tracing of inrush pathways in deep coal mines remain challenging, as elevated geothermal temperatures can alter the molecular fingerprints of coal-derived dissolved organic matter (Coal-DOM), a potential organic tracer. To address this, we investigated the evolution of Coal-DOM from coals of different ranks (long flame coal, lean coal, anthracite) under simulated geothermal conditions (25 and 50°C). By integrating ultrahigh-resolution mass spectrometry (FT-ICR MS) with an interpretable machine learning framework (XGBoost-SHAP and reactomics), we decoded the rank-specific molecular transformation pathways. Results showed that warming diversified the DOM pool from low-rank coals via fragmentation and oxidation, while it selectively enriched condensed aromatic and sulfur-containing structures in high-rank anthracite DOM, forming a stable and distinct fingerprint. Key molecular descriptors (O/C, NOSC, $AI_{mod}$, sulfur content) were identified as robust predictors of thermal reactivity. Overall, this integrated framework enables molecular-scale prediction of DOM reactivity in coal-bearing aquifers under geothermal perturbation. In addition, it yields quantifiable organic fingerprints that complement conventional indicators for mine-water source identification and water-inrush tracing. These capabilities can support environmental risk assessment and guide management of deep mine water systems.

## 1. Introduction

Coal has long played a fundamental role in the global energy mix. As shallow coal resources become increasingly depleted, mining activities are extending into deeper strata [1], where coal-bearing aquifers and mine-water systems are progressively exposed to geothermal conditions of 40–60 °C and even higher [2]. Under deep geothermal conditions, the physicochemical environment at the coal-water interface differs markedly from that at shallow, ambient conditions [3]. Hydrothermal processes can induce swelling, chain scission, structural rearrangement, and functional group

**Data availability statement:** All relevant data are within the paper and its Supporting Information files.

**Funding:** This work was supported by the National Key Research and Development Program of China (Grant No. 2024YFC3013803), awarded to Y.Q. Wu. The full name of the funder is the Ministry of Science and Technology of the People's Republic of China, and its official website is http://www.most.gov.cn/. The sponsors/funders had no role in the study design, data collection and analysis, decision to publish, or preparation of the manuscript.

**Competing interests:** The authors have declared that no competing interests exist.

transformation of the coal organic matrix [4,5], thereby continuously releasing dissolved organic matter (DOM) into the surrounding aquifer system. Coal-DOM constitutes a highly reactive organic carbon pool in groundwater and mine-water systems. As a core component of aquatic carbon cycling, DOM drives the biogeochemical dynamics of fluvial ecosystems, regulating microbial community assembly and nutrient turnover across river basins [6]. It not only participates in carbon cycling and redox buffering [7], but also influences the environmental behavior of metal(loid)s and organic contaminants through complexation and cotransport [7], thereby reshaping the water-quality evolution and risk profile of deep mine-water systems. Clarifying the molecular characteristics and thermal evolution mechanisms of Coal-DOM under geothermal conditions is therefore critical for ensuring the safety and resource-oriented utilization of deep coal mine water.

Coal rank reflects the degree of coalification and is a primary factor controlling the molecular structure of coal and its interfacial reactivity [8,9]. Low-rank coals are enriched in reactive oxygen-containing functional groups and labile aliphatic structures, and therefore tend to release larger amounts of more oxidized DOM [10,11]. In contrast, high-rank coals exhibit highly condensed aromatic frameworks and stable carbon skeletons, resulting in lower DOM yields and more inert molecular signatures [11,12]. Previous studies have largely relied on bulk indicators such as DOC, $UV_{254}$ and three-dimensional fluorescence (EEM), or on conventional structural parameters, to discuss coal-rank effects [13,14]. However, these approaches are insufficient to resolve the ultrahigh molecular complexity of Coal-DOM, particularly under geothermal conditions where multiple transformation pathways occur simultaneously, making it difficult for single metrics to establish robust structure-reactivity relationships. Ultrahigh-resolution Fourier transform ion cyclotron resonance mass spectrometry (FT-ICR MS) provides the core analytical basis for molecular-level characterization of DOM [13,15,16]. This technique can assign thousands of exact molecular formulas with sub-ppm mass accuracy, thereby enabling systematic characterization of key structural features of DOM, including elemental composition, heteroatom distribution, aromaticity, and oxidation state [17,18]. These molecular fingerprints form a fundamental basis for constraining the sources and reactivity of DOM. However, conventional formula-based statistical analyses primarily focus on compositional changes and are insufficient to resolve the specific chemical transformation pathways that occur under geothermal perturbation.

Reactomics uses paired mass distances (PMDs) as diagnostic clues, linking observed mass differences to plausible reaction types (e.g., oxygenation, dealkylation, chain cleavage, and decarboxylation) and thereby enabling the reconstruction of molecular-level reaction networks of DOM [19–21]. This approach has proven effective in disinfection and oxidation systems for identifying dominant transformation pathways [22,23]. However, the application of reactomics to the thermal evolution of Coal-DOM remains very limited, and the molecular "fate" during heating-namely, whether individual formulas are produced, removed, or remain unchanged-has not yet been systematically elucidated [24]. The introduction of explainable machine learning provides a new avenue to address this challenge. Tree-based models such as eXtreme Gradient Boosting (XGBoost) can be used to build "structure-fate" prediction models

in high-dimensional molecular descriptor space [25]. In combination with SHapley additive explanations (SHAP), model decisions can be decomposed into contributions from individual molecular features, allowing quantitative identification of key variables that govern the reactivity and transformation pathways of DOM [21,26,27]. Previous studies have shown that, in disinfection and advanced oxidation processes, parameters such as O/C, NOSC, $AI_{mod}$, molecular weight, and heteroatom content (e.g., N and S) are central to explaining the molecular reactivity of DOM [22,28,29]. Such an integrated framework that combines high-resolution mass spectrometry, reactomics, and explainable machine learning offers a powerful tool for extracting molecular fingerprints and reactive hotspots from complex DOM matrices. However, it has not yet been systematically applied to Coal-DOM systems controlled by the coupled effects of coal rank and geothermal conditions.

In this study, we selected three representative coal ranks–long flame coal (LFC), lean coal (LC), and anthracite (ANT)–and conducted 30-day coal-water reaction experiments at 25 °C (shallow reference) and 50 °C (geothermal simulation for deeper seams). By integrating FT-ICR MS based molecular formula assignment, PMD-reactomics, and XGBoost-SHAP explainable modeling, we systematically investigated the coupled effects of coal rank and geothermal conditions on the molecular fingerprints and thermal fate of Coal-DOM. The specific objectives of this study were to: (i) elucidate coal-rank-dependent differences in molecular composition and structural domains of Coal-DOM under the two temperature conditions; (ii) develop molecular descriptor-based machine learning models to fingerprint coal rank and temperature, and use SHAP to identify key discriminative features and potential threshold behaviors; and (iii) interpret coal-rank effects on DOM under geothermal perturbation from a molecular structure and reactivity perspective, with emphasis on the associated environmental implications. This work builds on the growing application of big data and machine learning in smart water management [30], providing transferable molecular-level evidence and a data-driven methodological framework for understanding the evolution of DOM in deep coal mine water and for assessing the coupled relationships among water chemistry, organic matter, and environmental risk in coal-bearing aquifers.

## 2. Materials and methods

### 2.1. Coal samples and preparation

Coal samples representing three distinct ranks petrographic ranks–long flame coal (LFC, bituminous coal), lean coal (LC, bituminous coal), and anthracite (ANT)–were obtained from Madiliang coalmine in the Inner Mongolia Autonomous Region, Dujiagou coalmine and Yuenan coalmine in the Shanxi Province, China. Coal-rank classification followed the standard GB/T 482–2008. Proximate analysis was conducted following GB/T 212–2008 to characterize basic physico-chemical properties, including moisture ($M_{ad}$), ash ($A_{ad}$), volatile matter ($V_{ad}$), and vitrinite reflectance ($R_0$%). Elemental contents of C, H, O, N, and S were determined using an Elementar Vario EL III analyzer (Germany) according to GB/T 476–2008 [31]. All analytical results are summarized in Table 1.

### 2.2 DOM leaching experiments

Batch leaching experiments were performed to simulate coal-water interactions under contrasting geothermal conditions. For each treatment, 200.0 g of pretreated coal was combined with 2,000 mL of ultrapure water in a 2,500 mL acid-washed amber glass bottle. The suspensions were incubated in the dark at 25 or 50 °C with constant agitation (120 rpm) for 30 d.

Table 1. Industrial analysis and element analysis of coal samples.

| Coal sample | Industrial analysis (%) | | | Elemental analysis (%) | | | | | $R_0$ (%) |
|---|---|---|---|---|---|---|---|---|---|
| | $M_{ad}$ | $A_{ad}$ | $V_{ad}$ | C | H | O | N | S | |
| LFC | 5.36 | 19.93 | 29.43 | 64.25 | 3.74 | 12.53 | 1.59 | 0.33 | 0.615 |
| LC | 1.24 | 8.90 | 16.37 | 87.37 | 3.84 | 5.63 | 1.72 | 0.45 | 1.831 |
| ANT | 4.01 | 11.03 | 6.42 | 81.78 | 2.29 | 3.45 | 1.66 | 0.26 | 3.13 |

After incubation, the suspensions were filtered through 0.45 µm glass fiber filters. The retained solids were collected for subsequent solid-phase characterization, and the filtrates were stored at 4 °C in pre-cleaned glass vials for DOM analysis. All treatments were conducted in triplicate to ensure reproducibility. It should be noted that our study focuses on the characteristic molecular transformation pathways and identifiable fingerprints of Coal-DOM from different ranks under thermal perturbation, rather than long-term geological-scale evolution (hundreds to thousands of years). The 30-day laboratory simulation can sufficiently amplify the differential response of Coal-DOM from different ranks to geothermal conditions, and extract stable molecular fingerprints for mine water source tracing, fully meeting the research objectives.

## 2.3. FT-ICR MS analysis of Coal-DOM

The molecular evolution of Coal-DOM was monitored using a Fourier transform ion cyclotron resonance mass spectrometer (FT-ICR MS, SolariX XR 15T, Bruker, USA) equipped with a negative electrospray ionization (ESI) source. To ensure analytical robustness and data integrity, the following procedures were implemented: [32]. (1) Sample preparation: Each filtered sample was passed through a 0.22 µm membrane, acidified to pH = 2 with formic acid, enriched by solid-phase extraction on a polymeric reversed-phase sorbent, eluted with methanol, gently evaporated to near dryness under a stream of nitrogen, and finally reconstituted in methanol/water (1:1, v/v). (2) Instrument configuration and calibration: The optimized FT-ICR MS parameters were as follows: injection volume 200 µL, flow rate 120 µL h$^{-1}$, capillary voltage 4.0 kV, ion accumulation time 0.06 s, and mass-to-charge (m/z) range 100–1600 Da. Mass accuracy was ensured through a two-step calibration procedure. First, external calibration was performed using a 10 mM sodium formate solution prior to sample analysis. Subsequently, internal calibration was conducted with a set of reference organic compounds with well-established molecular formulas. This combined calibration procedure achieved a mass error of better than 1 ppm. (3) Data processing: To eliminate background interferences, any peaks detected in the procedural blanks and matching sample peaks within the predefined mass tolerance were removed from the final dataset. When multiple candidate formulas matched the same m/z value, the formula with the fewest heteroatoms and/or the smallest mass error was selected. Downstream analyses were restricted to an m/z range of 100–800 Da to ensure optimal mass resolution and detection reliability. Specifically, the double bond equivalent (DBE) was calculated to quantify molecular unsaturation, the nominal oxidation state of carbon (NOSC) was used to characterize redox properties, and the modified aromaticity index (AI$_{mod}$) was employed to evaluate the degree of aromatic condensation. $DBE = C - H/2 + N/2 + 1$; $NOSC = 4 - (4C + H - 3N - 2O - 2S)/C$; $AI = [1 + C - 0.5O - S - 0.5(N + H)]/(C - 0.5O - S - N)$; $AI_{mod}$, $w = \Sigma AI_{mod} \cdot I(m/z)/\Sigma I(m/z)$ [33,34]. Molecular formulas of DOM detected by FT-ICR MS were classified into seven compositional classes [22]: aliphatic/protein-like, lipid-like, lignin/CRAM-like, unsaturated hydrocarbon-like, carbohydrate-like, aromatic, and tannin-like structures. Aliphatic/proteins: $1.5 \leq H/C \leq 2.2$, $0.3 \leq O/C \leq 0.67$; Lipids: $1.5 \leq H/C \leq 2.0$, $0 \leq O/C \leq 0.3$; Lignins/CRAM-like structures: $0.7 \leq H/C \leq 1.5$, $0.1 \leq O/C \leq 0.67$; Unsaturated hydrocarbons: $0.7 \leq H/C \leq 1.5$, $0 \leq O/C \leq 0.1$; Carbohydrates: $1.5 \leq H/C \leq 2.4$, $0.67 \leq O/C \leq 1.2$; Aromatic structures: $0.2 \leq H/C \leq 0.7$, $0 \leq O/C \leq 0.67$; Tannins: $0.6 \leq H/C \leq 1.5$, $0.67 \leq O/C \leq 1.2$ [15].

## 2.4. Machine learning analysis

The XGBoost-SHAP framework was selected for this study for two core reasons: (1) it delivers superior robustness on our high-dimensional, sparse FT-ICR MS dataset compared to other ensemble methods (e.g., random forest), via built-in regularization to avoid overfitting; (2) its optimized compatibility with SHAP TreeExplainer enables fully interpretable quantification of molecular feature contributions, which aligns with our core goal of identifying key descriptors governing DOM thermal reactivity. This framework advances traditional bulk-scale contaminant fate models [35]. This analytical framework was built on a carefully curated three-class labeled dataset (classes 0, 1, and 2) to represent DOM transformation along three dimensions: fate (removed, unchanged, and produced), molecular category (e.g., lignin/CRAM-like, tannin-like, and aromatic structures), and elemental class (e.g., CHO, CHON, and CHOS). Training and test sets used an identical feature

representation to ensure comparability. For multiclass classification, we trained an XGBoost model using the multi:softmax objective (num_class = 3) and optimized performance with multinomial log loss (mlogloss) as the primary metric. Prior to training, hyperparameters were tuned via automated grid search with 10-fold cross-validation, focusing on the learning rate (η), maximum tree depth (max_depth), and minimum child weight (min_child_weight). The optimal hyperparameter set was selected by minimizing cross-validated loss to improve model robustness. The final model was evaluated on a held-out test set to quantify predictive performance. Model interpretability was assessed post hoc using SHapley additive explanations (SHAP) to quantify the contribution of each feature to class predictions [24].

## 3. Results and discussion

### 3.1. Molecular fingerprints of Coal-DOM under geothermal conditions

FT-ICR MS showed that Coal-DOM from different coal ranks exhibited highly complex molecular compositions under both 25 °C and 50 °C conditions. The weighted average molecular metrics for all samples, including $O/C_w$, $H/C_w$, $DBE_w$, $AI_w$, and $NOSC_w$, are summarized in Table 2. Van Krevelen diagrams (Fig 1) indicated that most molecular formulas were located within an O/C range of 0.10–0.60 and an H/C range of 0.70–1.80, corresponding to partially oxidized aromatic–aliphatic mixtures dominated by lignin/CRAM-like and tannin-like structures, while a smaller fraction of formulas extended into a high-O/C, high-H/C region characteristic of carbohydrate-like compounds and into a low-H/C region associated with condensed aromatic structures. When grouped by elemental class, CHONS formulas formed a dense core cluster in the intermediate O/C–H/C space, whereas CHOS formulas were enriched in the high-O/C, high-H/C domain. By comparison, CHO and CHON formulas displayed a broader, fan-shaped spread across H/C, underscoring substantial variability in heteroatom content and saturation within Coal-DOM.

Warming from 25 to 50 °C caused the molecular point clouds for all coal ranks to expand toward higher O/C and H/C. CHOS and CHONS formulas increased markedly in the high-H/C domain, and a subset of CHO/CHON formulas shifted toward higher O/C. These patterns suggest that geothermal heating enhances cleavage and oxygenation of aliphatic moieties and promotes dissociation and rearrangement of N- and S-bearing functional groups, which may further influence the biogeochemical cycling of nitrogen and sulfur in mine-influenced aquatic systems [36], collectively steering DOM toward more oxidized and more saturated structures [37, 38].

Coal rank markedly modulated the thermal response. LFC-DOM showed the broadest spread in O/C-H/C space and pronounced enrichment of CHOS and CHONS formulas at high H/C, whereas LC-DOM exhibited a more constrained distribution. In contrast, ANT-DOM was shifted toward lower O/C and H/C and displayed the smallest temperature-induced change, consistent with dominance of stable condensed aromatic frameworks and relative insensitivity to short-term heating [39]. Collectively, coal rank sets the baseline structural domains and heteroatom patterns of Coal-DOM, while geothermal warming accentuates these contrasts by redistributing formulas along the O/C and H/C axes. This rank- and temperature-dependent shift establishes a molecular context for interpreting subsequent reactivity analyses.

**Table 2. A summary of the average molecular metrics of Coal-DOM under various treatments, as determined by FT-ICR MS.**

| Group | Molecular Number | CHO | CHON | CHOS | CHONS | $H/C_w$ | $O/C_w$ | $DBE_w$ | $NOSC_w$ | $M/Z_w$ | $AI_w$ |
|---|---|---|---|---|---|---|---|---|---|---|---|
| LFC-DOM-25 °C | 4675 | 2235 | 1473 | 643 | 324 | 1.493 | 0.303 | 6.099 | −0.817 | 358 | 1.807 |
| LC-DOM-25 °C | 3877 | 1996 | 889 | 679 | 313 | 1.456 | 0.313 | 6.553 | −0.771 | 363 | 0.182 |
| ANT-DOM-25 °C | 4256 | 1466 | 1364 | 571 | 855 | 1.636 | 0.234 | 4.919 | −1.037 | 355 | 0.087 |
| LFC-DOM-50 °C | 6840 | 3318 | 2300 | 940 | 282 | 1.287 | 0.402 | 8.236 | −0.439 | 405 | 0.262 |
| LC-DOM-50 °C | 7876 | 3777 | 1855 | 1668 | 576 | 1.279 | 0.390 | 9.043 | −0.448 | 440 | 0.266 |
| ANT-DOM-50 °C | 7476 | 3142 | 3348 | 709 | 277 | 1.685 | 0.261 | 4.857 | −1.099 | 437 | 0.095 |

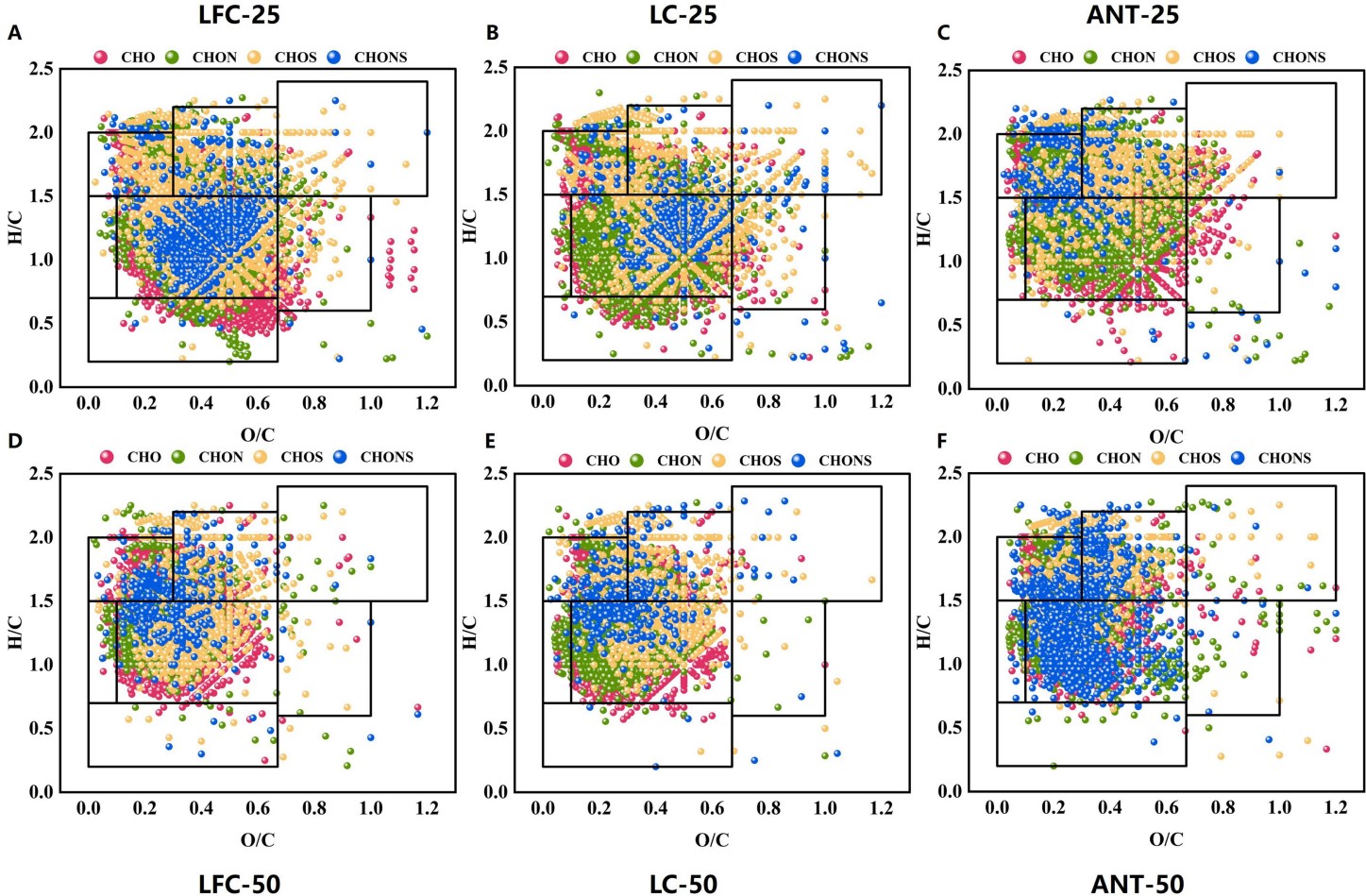

**Fig. 1. Van Krevelen diagrams illustrating the molecular composition of Coal-DOM from LFC (A, D), LC (B, E), an ANT (C, F) under different geothermal conditions.** Molecular formulas are color-coded by elemental classes (CHO, CHON, CHOS, and CHONS).

To characterize temperature-driven molecular fate, formulas detected only at 25 °C were classified as removed, those detected only at 50 °C as produced, and those present at both temperatures as resistant (unchanged). In the two-dimensional NOSC versus (DBE–O)/C space (Fig 2A-C), all Coal-DOM systems displayed a spindle-shaped distribution, yet the three fate classes occupied distinct regions. Removed formulas clustered at lower NOSC (< 0) and intermediate (DBE–O)/C, consistent with preferential depletion of relatively reduced structures with moderate aromaticity upon warming. Produced formulas were shifted toward higher NOSC and, in some cases, slightly lower (DBE–O)/C, indicating that 50 °C favored the emergence of more oxidized and less condensed carbon skeletons. By contrast, resistant formulas concentrated within a narrow band (NOSC ≈ −0.5 to 0.5; (DBE–O)/C ≈ 0), representing a background pool of moderately oxidized structures with low-to-moderate aromaticity that remained comparatively stable under thermal perturbation. The distribution of structural domains (Fig 2D) was highly consistent with these redox patterns. Across all samples, carbohydrate-like formulas contributed the largest fraction, followed by lignin/CRAM-like and tannin-like structures; lipid-/protein-like and highly aromatic domains were minor. Together, this domain structure indicates that Coal-DOM is dominated by partially oxidized, weakly condensed components. At 25 °C, ANT-DOM exhibited a lower fraction of carbohydrate-like formulas and higher fractions of tannin-like and aromatic structures than LFC-DOM and LC-DOM,

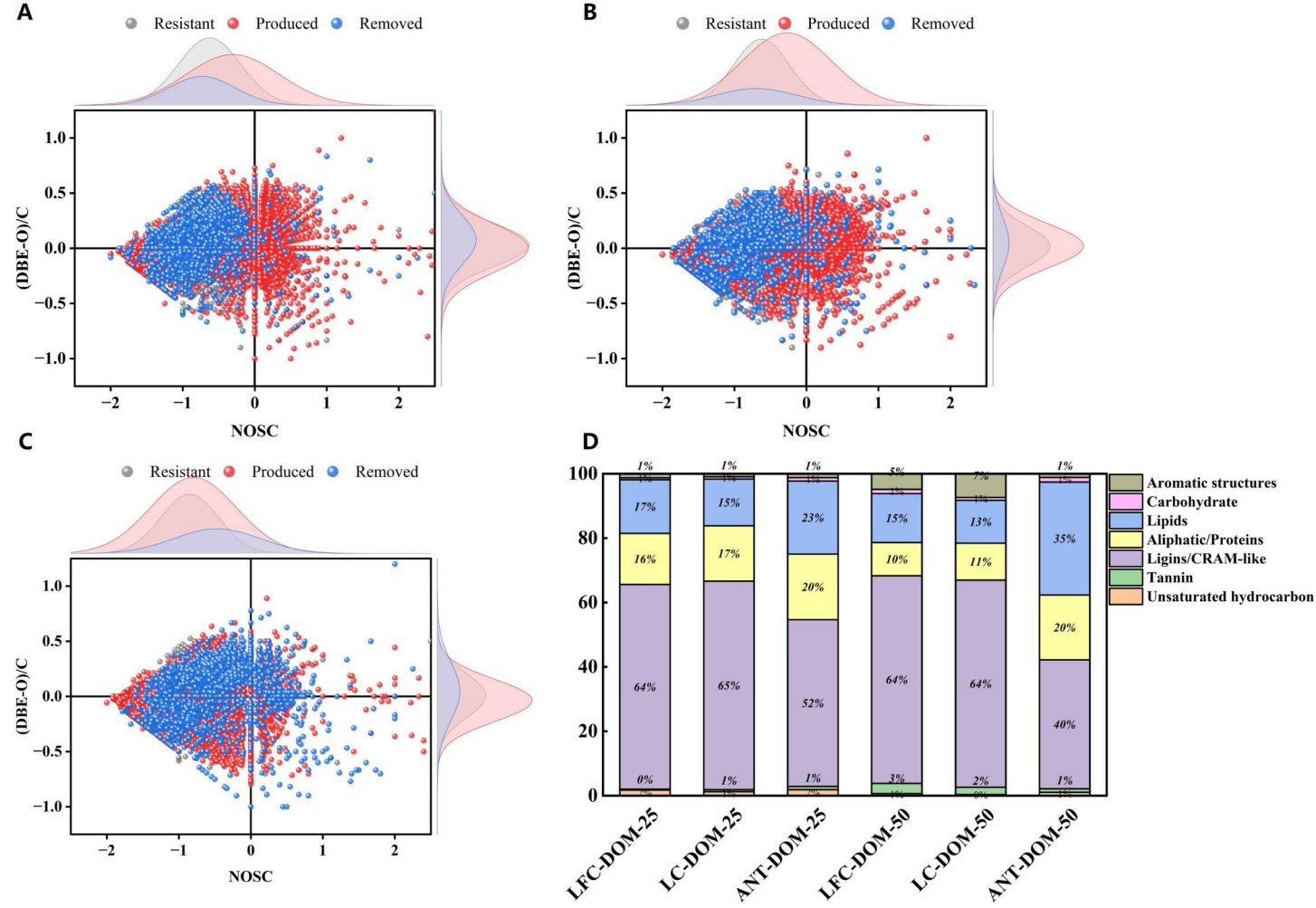

**Fig 2. The marginal density peaks in the (DBE–O)/C versus NOSC plots illustrate the abundance distributions of these two parameters for Coal-DOM components, (including precursor, resistant, and newly produced molecules), under different geothermal conditions.**

indicating a more aromatic and condensed baseline signature for high-rank coal-DOM. Upon heating to 50 °C, the relative contribution of carbohydrate-like formulas changed little in LFC- and LC-DOM, with only slight increases in lipid-/protein-like and some aromatic structures. In contrast, ANT-DOM showed a pronounced decrease in carbohydrate-like formulas accompanied by marked increases in tannin-like and aromatic structures, indicating selective enrichment of stable condensed frameworks in high-rank coal [40]. Overall, these patterns suggest divergent thermal responses: low-rank Coal-DOM is more susceptible to oxygenation and partial cleavage of oxygen-rich aliphatic segments, whereas high-rank DOM is dominated by mild restructuring and preferential enrichment of pre-existing aromatic/CRAM-like backbones.

### 3.2. Transformation pathways of Coal-DOM

To elucidate the underlying transformation pathways at the molecular level, performed mass difference (PMD) analysis was conducted on Coal-DOM. Fig 3 shows that thermally induced molecular transformations could be grouped into five major types: oxygenation, cleavage, dealkylation, carboxyl-related reactions, and other processes [41]. Overall, oxygenation events ($+O$, $+O+2H$), cleavage reactions ($-C_nH_{2n+2}O_m$), and carboxyl-related losses ($-CO_2$, $-C_2H_2O_2$) exhibited

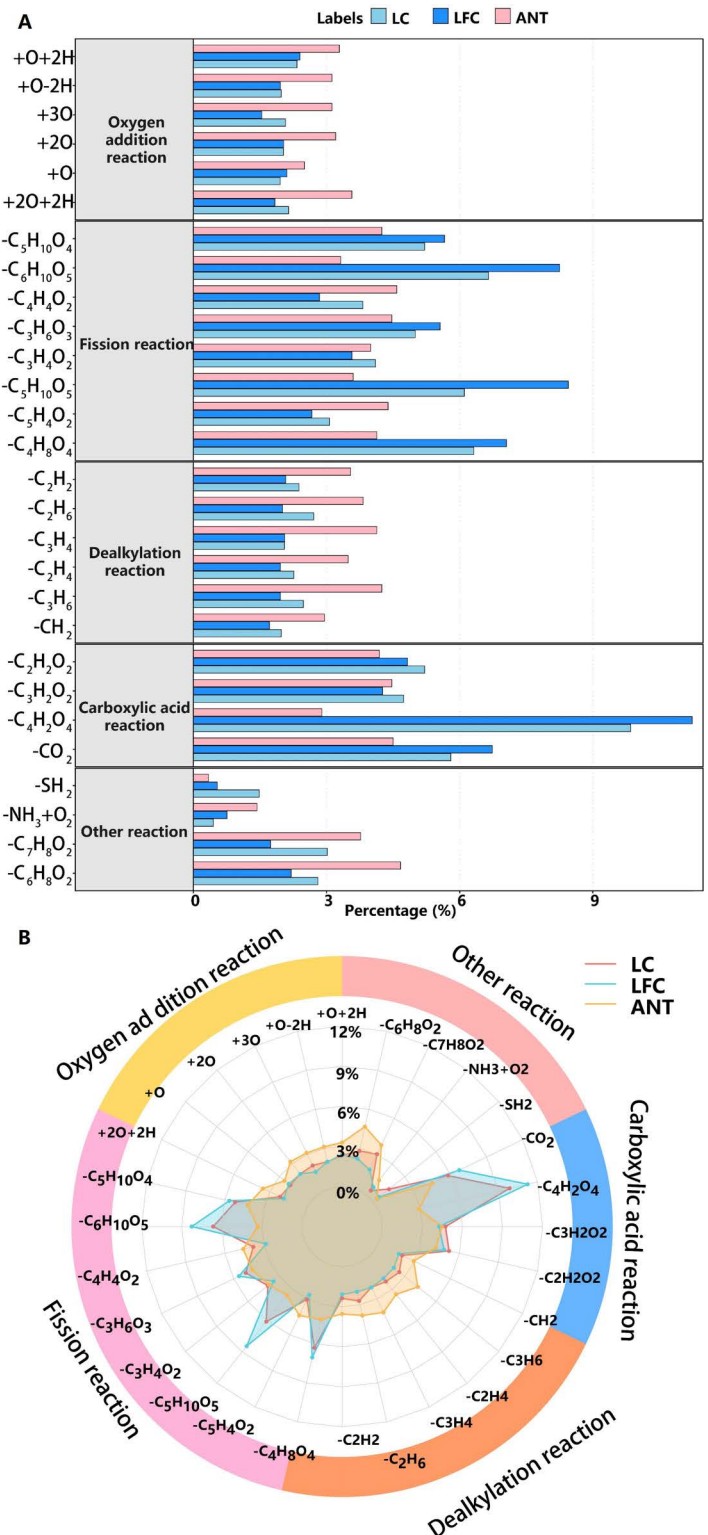

**Fig 3. (A) Reaction types and (B) a radar chart summarizing the conversion pathways inferred from precursor–product pairs in Coal-DOM across temperatures.**

much higher relative abundances than dealkylation and other pathways, indicating that at 50°C the evolution of Coal-DOM is primarily governed by the introduction and rearrangement of oxygenated functional groups, the breakdown of oxygen-rich chain segments, and the removal of carboxylic/carboxylate moieties. Marked differences in reaction spectra were observed among coal ranks. In LFC-DOM, cleavage-type PMDs such as $-C_5H_{10}O_4$ and $-C_4H_8O_4$, together with carboxyl-related PMDs, contributed the largest shares, indicating that low-rank coal is more prone to extensive depolymerization and decarboxylation of hydroxyl- and carboxyl-rich aliphatic chains during heating, and thus represents a major source of low-molecular-weight, high-NOSC products [42]. LC-DOM still exhibited a pronounced signature of cleavage reactions, but the relative contributions of oxygenation and dealkylation were comparable to those of cleavage. In contrast, ANT-DOM was characterized by relatively higher proportions of dealkylation and oxygenation reactions and only minor contributions from cleavage and carboxyl-related pathways, indicating that, within the temperature range investigated, high-rank coal primarily experiences removal of alkyl side chains from aromatic cores and slight oxygenation of the backbone, while extensive breakdown of the pre-existing condensed aromatic framework is unlikely to occur. Collectively, low-rank Coal-DOM is dominated by deep depolymerization driven by cleavage and decarboxylation, whereas high-rank DOM mainly undergoes backbone refinement through dealkylation and oxygenation; LC-DOM displays an intermediate transformation regime.

### 3.3. Coupled controls of molecular domains, descriptors and thermal reactivity

To elucidate the overall coupling between "molecular domains-traits-fate", a Sankey diagram (Fig 4) was constructed using elemental/structural classes, key molecular descriptors, and molecular fate (precursor, product, and resistant) as nodes. From an elemental-composition perspective, CHON and CHONS formulas preferentially flowed to the product node through channels characterized by high NOSC, relatively high (DBE–O)/C, and medium-to-high $AI_{mod}$, indicating that relatively oxidized, aromatic–unsaturated N- and S-containing structures are more reactive at 50 °C. In contrast, formulas dominated by CHO with low NOSC and (DBE–O)/C values close to zero mainly connected precursor and resistant nodes, representing an inert background pool of low-oxidation, moderately unsaturated molecules. CHOS formulas showed a more dispersed

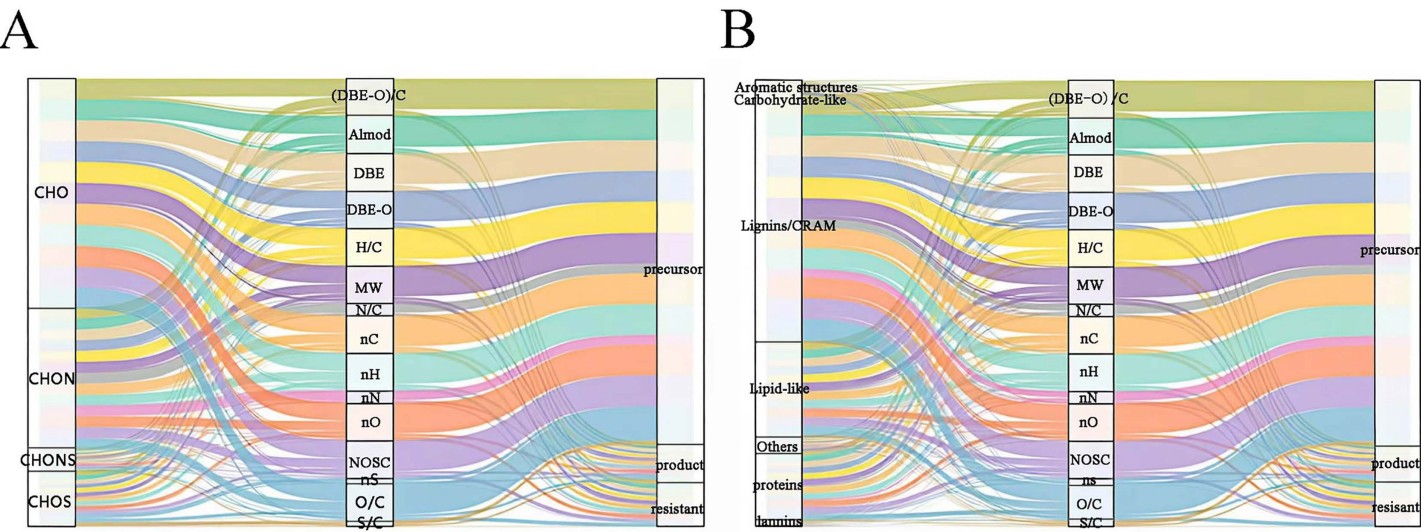

**Fig 4. Sankey diagrams illustrating (A) elemental composition and (B) molecular categories, constructed using normalized SHAP values.**

behavior: one fraction was routed to the product node via pathways with high O/C and elevated NOSC, whereas another fraction linked to the resistant node via lower O/C and DBE values, suggesting that S-containing structures comprise both highly reactive and relatively stable components [43,44]. A structural-domain perspective further refined these patterns. Carbohydrate-like formulas mainly flowed to precursor and resistant nodes via pathways with low $AI_{mod}$, relatively high H/C, and intermediate molecular weight (MW), suggesting that they undergo only limited oxygenation or cleavage over the 30-day thermal perturbation. In contrast, lignin/CRAM-like and tannin-like formulas were strongly connected to the product node through combinations of higher $AI_{mod}$, medium-to-high DBE, and elevated NOSC, indicating that warming enhances selective oxygenation and partial rearrangement of these aromatics-dominated structures, which are key sources of newly formed high-NOSC molecules [15,45]. Lipid-like and protein/amino sugar-like formulas were split within a region of high H/C, low $AI_{mod}$, and intermediate DBE-O, acting both as precursors that undergo cleavage to yield smaller products and as intermediates that can form more polar species through coupled oxygenation–dealkylation processes. Overall, the Sankey results are highly consistent with the PMD-reactomics-based reaction channels, demonstrating that N/S-containing aromatic–unsaturated structures constitute the most readily activated "reactive core" under geothermal heating, whereas CHO-dominated, low-oxidation, low-to-moderate aromatic structures form a thermally stable background.

### 3.4. ML analysis of the molecular characteristics, composition, and reactivity of Coal-DOM

The integration of machine learning with FT-ICR MS provides a robust approach for deciphering the complex molecular composition and transformation dynamics of DOM. Novel machine learning frameworks have been proven effective for modeling the fate and transport of complex organic matter in aquatic environments [35]. The XGBoost-based three-class model ("produced-removed-unchanged") performed well overall, indicating that DOM molecular traits can effectively fingerprint temperature-induced molecular fate. Global SHAP analysis (Fig 5) showed that O/C, NOSC, $AI_{mod}$, and molecular weight (MW) were consistently the most important features across all three Coal-DOM systems, followed by H/C and the numbers of heteroatoms (nN, nO, nS), whereas DBE, (DBE–O)/C, and nC contributed relatively little to model predictions. These results suggest that the partitioning of molecules into "produced-removed-unchanged" groups with increasing temperature is governed by the combined effects of oxidation state, degree of aromatic condensation, molecular weight, and N/S enrichment, rather than by any single structural descriptor. Examining SHAP patterns by fate class in low-rank LFC-DOM and LC-DOM, O/C and NOSC showed markedly higher SHAP values for the produced class than for the removed and unchanged classes, indicating that high-O/C, high-NOSC molecules are most likely to be newly formed or enriched at 50 °C [46]. In contrast, $AI_{mod}$ and MW contributed slightly more to the removed class, suggesting that highly aromatic, high-molecular-weight structures tend to be preferentially consumed or transformed during heating [47]. For the unchanged class, SHAP values for most descriptors were intermediate or low, consistent with the previously identified "moderate-oxidation, moderate-unsaturation" stability domain (NOSC ≈ −0.5–0.5, (DBE-O)/C ≈ 0). In ANT-DOM, SHAP values for $AI_{mod}$, MW, and nS were substantially higher than those for other descriptors and peaked in the unchanged class, indicating that highly aromatic, high-molecular-weight, S-containing structures are key traits underlying the thermal inertness of high-rank Coal-DOM between 25 and 50 °C. At the same time, O/C and NOSC still showed elevated contributions for produced molecules, suggesting that oxygen addition and increases in oxidation state remain the primary drivers of new product formation even in strongly condensed aromatic systems. Notably, the core sulfur-containing structures in ANT-DOM remain stable and identifiable even under geothermal warming, making them ideal persistent molecular tracers for water sources influenced by high-rank coal seams. In addition, these organosulfur compounds may mediate the complexation and mobility of heavy metals in deep mine water, which has direct implications for assessing the long-term environmental risk of deep mining activities.

To obtain more intuitive threshold information in the continuous feature space, SHAP dependence plots of the XGBoost models for LFC-DOM and LC-DOM were generated for O/C-NOSC, NOSC-$AI_{mod}$, and O/C-$AI_{mod}$ (Fig 6,7). For LFC-DOM, high SHAP values for the produced class were concentrated in the quadrant with O/C > 0.4 and NOSC > 0, whereas the removed class formed a band of high contributions at O/C ≈ 0.2–0.4 and NOSC < 0. Unchanged formulas clustered within

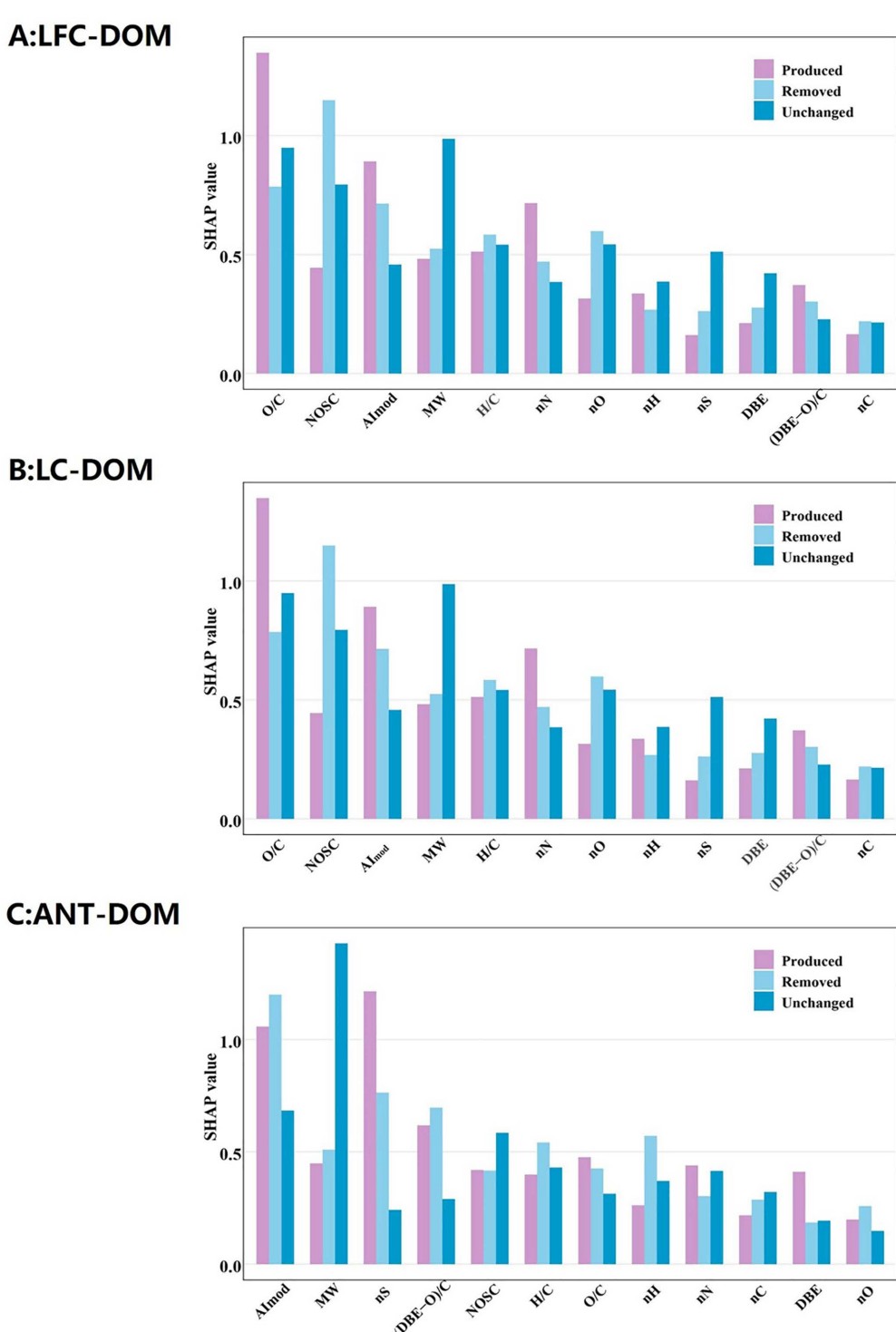

**Fig 5. SHAP analysis ranked the contributions of Coal-DOM molecular characteristics to reaction activity(A-C).** (MW, molecular weight; nN, number of N atoms; nS, number of S atoms; nH, number of H atoms; nO, number of O atoms; nC, number of C atoms).

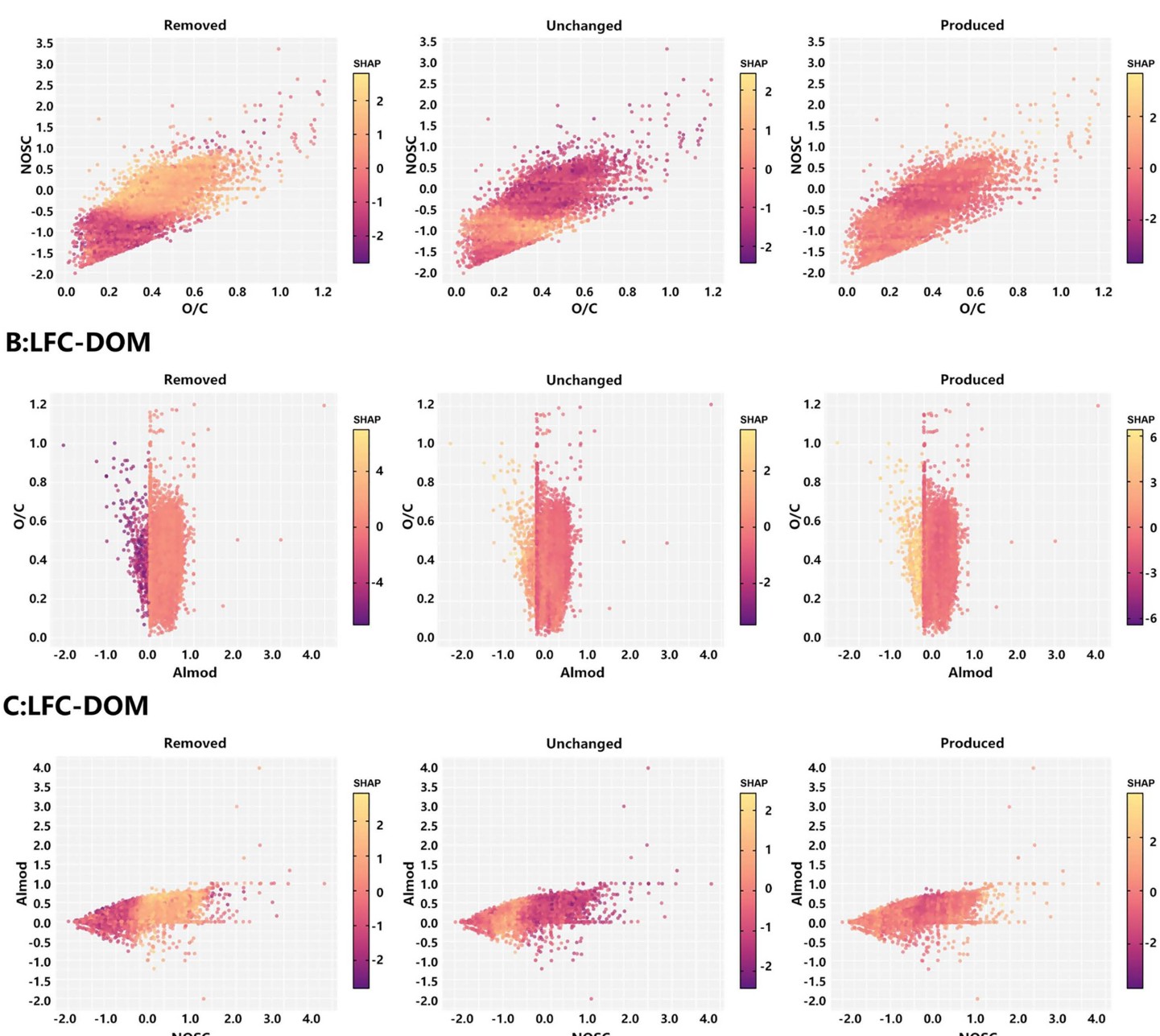

**Fig 6. Combined shape feature analysis of molecular characteristics and reactivity in LFC-DOM: (A) O/C ratio vs. NOSC, (B) $AI_{mod}$ vs. O/C, (C) NOSC vs. $AI_{mod}$.**

a narrow domain of O/C ≈ 0.3–0.5 and NOSC ≈ −0.5–0.5, with overall low SHAP values. These patterns indicate that, for LFC-DOM, the high-O/C, high-NOSC region represents a thermal "hotspot" for activation, whereas relatively reduced structures with intermediate O/C and negative NOSC are preferentially consumed, and molecules with intermediate

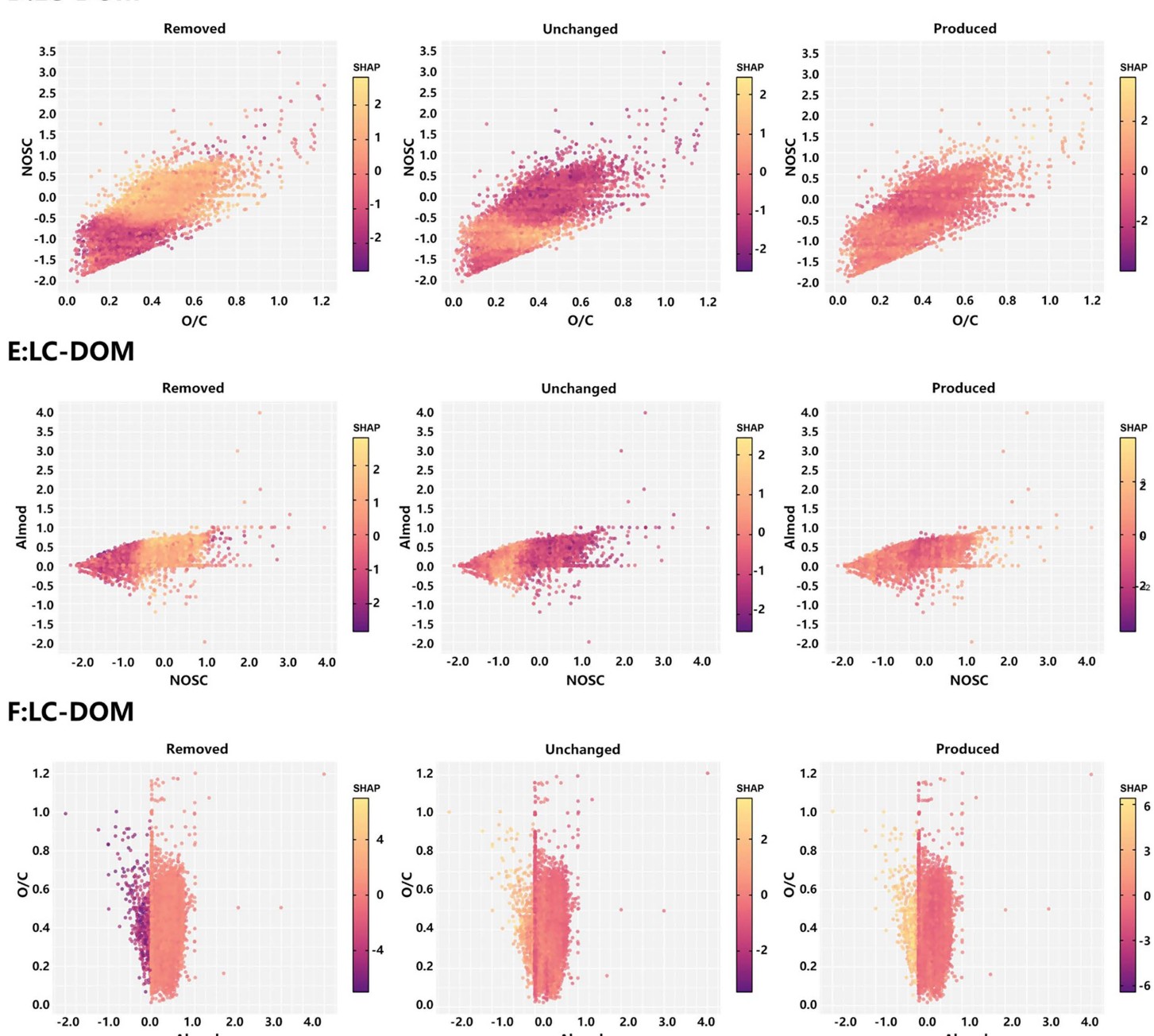

**Fig 7. Combined shape feature analysis of molecular characteristics and reactivity in LC-DOM: (A) O/C ratio vs. NOSC, (B) NOSC vs. AI$_{mod}$, (C) AI$_{mod}$ vs. O/C.**

oxidation states constitute a stable background under thermal perturbation. In the NOSC-AI$_{mod}$ and O/C-AI$_{mod}$ planes, high SHAP regions for produced formulas in LFC-DOM were mainly located at NOSC > 0 with AI$_{mod}$ ≈ 0–0.5, or at O/C > 0.4 with intermediate AI$_{mod}$, indicating that moderately aromatic, more oxidized structures are the primary carriers of thermal

activation in LFC-DOM. By contrast, removed formulas were shifted toward regions with $AI_{mod} > 0.5$ and NOSC that is moderate or slightly negative, suggesting that a subset of condensed aromatic frameworks undergo preferential rearrangement or cleavage upon heating. The SHAP dependence patterns of LC-DOM (Fig 7) were broadly similar to those of LFC-DOM, but the regions of high contribution were markedly narrower. For produced formulas, high SHAP values were confined to a narrow window with $O/C \approx 0.35–0.55$, $NOSC > 0$, and $AI_{mod} \approx 0–0.4$, while high-SHAP areas for the removed class were smaller than in LFC-DOM. This indicates that, with increasing coal rank, the thermal activation window narrows and the proportion of moderately oxidized aromatic structures that can be preferentially consumed decreases, and that the evolution of LC-DOM is more biased toward mild oxygenation and localized side-chain modification.

For ANT-DOM, SHAP dependence patterns were further examined in three feature planes: $MW\text{-}AI_{mod}$, MW-nS, and $nS\text{-}AI_{mod}$ (Fig 8). The $MW\text{-}AI_{mod}$ plane showed that ANT-DOM formulas were mainly confined to a narrow region with $MW \approx 200–800$ Da and $AI_{mod} \approx 0–0.6$. SHAP values for produced and removed formulas differed only moderately, with slight separation occurring in the intermediate molecular-weight range (~300–500 Da), indicating that, in high-rank coal systems, molecular weight and aromaticity provide limited discriminatory power for thermal fate. By contrast, the MW-nS and $nS\text{-}AI_{mod}$ plots clearly revealed the dominant control of sulfur content on the thermal reactivity of ANT-DOM. Across the entire MW range, produced formulas with $nS = 1$ exhibited substantially higher SHAP values than those with $nS = 0$, whereas for the removed and unchanged classes, SHAP values for $nS = 1$ were mostly negative or close to zero. This pattern indicates that S-containing structures constitute the principal reactive pool of ANT-DOM, while S-free aromatic backbones remain largely inert within the 25–50 °C temperature window. The $nS\text{-}AI_{mod}$ plot further showed that produced formulas with $nS = 1$ maintained consistently high positive SHAP values across almost the entire $AI_{mod}$ range (0–0.6), whereas formulas with $nS = 0$ contributed very little regardless of changes in aromaticity. Overall, these results demonstrate that, in the highly aromatic ANT-DOM system, the presence of sulfur atoms itself acts as a key "activation trigger", whereas subtle variations in $AI_{mod}$ and MW only modulate molecular fate to a secondary extent.

### 3.5. Implications for environmental management

This study provide molecular-level insights with direct implications for managing water resources and risks in deep mining environments. Firstly, the identified rank-specific molecular fingerprints (e.g., the dominance of aliphatic compounds in anthracite DOM vs. oxidized, heteroatom-rich molecules in low-rank coal DOM) enable high-precision identification of groundwater sources affected by coal-water interactions. This enhances the accuracy of early warning systems for water-inrush disasters. Secondly, the temperature-driven evolution of DOM molecular reactivity (e.g., increased oxygenation in low-rank coals) helps predict the potential of Coal-DOM to mobilize co-occurring pollutants like heavy metals, which is crucial for assessing the long-term environmental risk of mine water. Therefore, we recommend incorporating the characteristic molecular ratios (e.g., O/C, $AI_{mod}$) identified by this study into future monitoring protocols for coal-bearing aquifers. Combined with emerging technologies such as hyperspectral remote sensing for real-time water quality monitoring [48], these molecular fingerprints can be integrated into big data-driven smart water management systems [30], moving beyond traditional inorganic ions to build a more robust source-tracing and risk assessment framework for deep mine water. It should be noted that this study used a 30-day laboratory incubation to simulate the initial water-coal interaction after mining excavation. Future studies can adopt longer incubation periods to simulate the long-term geological-scale evolution of Coal-DOM in deep aquifers, to further improve the applicability of the molecular fingerprint framework for long-term mine water management.

### 4. Conclusions

This study integrates FT-ICR MS, reactomics, and explainable machine learning (XGBoost-SHAP) to decipher the coal-rank-dependent evolution of molecular fingerprints in Coal-DOM under simulated geothermal conditions (25–50 °C), constructing a diagnostic framework that links molecular signatures to environmental management. The results

## G:ANT-DOM

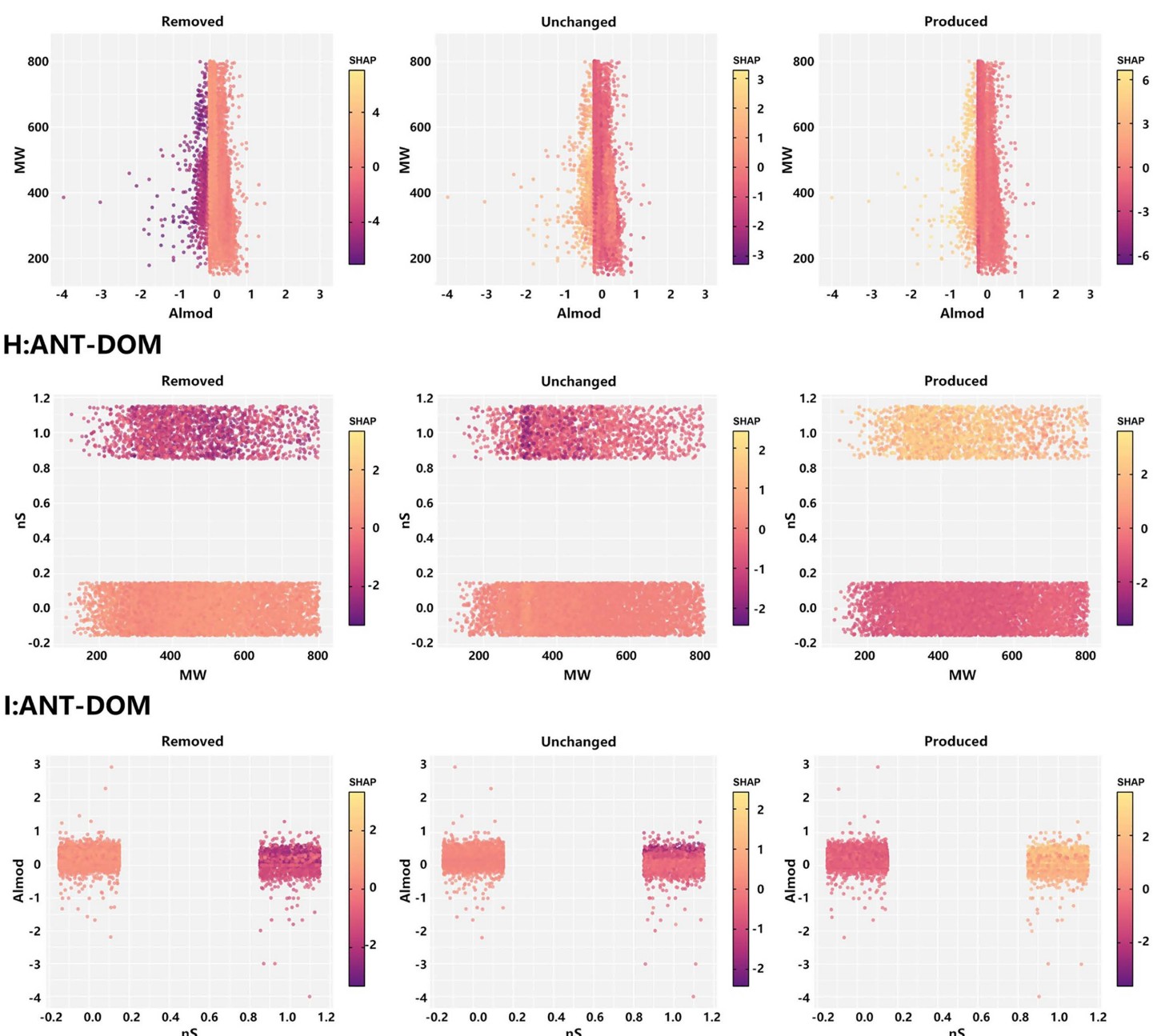

## H:ANT-DOM

## I:ANT-DOM

**Fig 8. Combined shape feature analysis of molecular characteristics and reactivity in LC-DOM: (A) $AI_{mod}$ vs. MW, (B) MW vs. nS, (C) nS vs. $AI_{mod}$.**

demonstrate that elevated temperature amplifies the divergence of DOM released from different coal ranks: low-rank coal DOM becomes more oxidized and diverse through fragmentation and oxidation pathways, while high-rank anthracite DOM remains stable and forms a unique fingerprint enriched with aromatic structures and sulfur-containing compounds. Machine-learning models further identify key molecular descriptors-O/C, NOSC, $AI_{mod}$, and sulfur content, as

robust predictors of thermal reactivity. These predictable, rank-specific molecular fingerprints can serve as a novel organic tracing tool that complements conventional hydrogeochemical methods, directly supporting precise source tracking and management of deep mine water resources. Specifically, this tool enhances the discrimination of water sources from coal-bearing aquifers, assists in modeling the influence of geothermal gradients on inrush-water composition to trace flow pathways, and guides priority monitoring of contamination-risk hotspots through the identification of reactive DOM components. The establishment of this molecular diagnostic framework provides environmental managers with evidence-based decision-support, contributing to improved water-hazard prevention and the sustainable management of water resources in deep coal-mining areas.

## Supporting information

**S1 File. Supplementary tables and raw data.**
(ZIP)

## Author contributions

**Investigation:** Peng Ge, Tan Liu, Dong Dong, Zepeng Wan, Yanqing Wu.

**Methodology:** Yanqing Wu.

**Supervision:** Yanqing Wu.

**Writing – original draft:** Peng Ge, Zepeng Wan.

**Writing – review & editing:** Peng Ge, Zepeng Wan, Yanqing Wu.

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
