## [Decision Letter · Decision Letter 0]

17 Mar 2026

PONE-D-26-04852Molecular signature evolution of coal-derived dissolved organic matter under geothermal conditions: FT-ICR MS and machine learningPLOS One

Dear Dr. Wan,

Thank you for submitting your manuscript to PLOS ONE. After careful consideration, we feel that it has merit but does not fully meet PLOS ONE’s publication criteria as it currently stands. Therefore, we invite you to submit a revised version of the manuscript that addresses the points raised during the review process.

We look forward to receiving your revised manuscript.

Kind regards,

Muammar Qadafi

Academic Editor

PLOS One

**Journal Requirements:**

6. Thank you for stating the following in the Competing Interests section:

“The authors have declared that no competing interests exist.”

We note that one or more of the authors are employed by a commercial company: “China Coal Technology & Engineering Group Hangzhou Research Institute Co., Ltd.”

7. We note you have included a table to which you do not refer in the text of your manuscript. Please ensure that you refer to Table 2 in your text; if accepted, production will need this reference to link the reader to the Table.

**Additional Editor Comments:**

Please note that the authors are not required to cite the reference(s) suggested by the reviewer unless they are essential and directly relevant to the study.

Reviewers' comments:

Reviewer's Responses to Questions

**Comments to the Author**

1. Is the manuscript technically sound, and do the data support the conclusions?

Reviewer #1: Yes

Reviewer #2: Yes

2. Has the statistical analysis been performed appropriately and rigorously? 

Reviewer #1: Yes

Reviewer #2: Yes

3. Have the authors made all data underlying the findings in their manuscript fully available?

Reviewer #1: Yes

Reviewer #2: Yes

4. Is the manuscript presented in an intelligible fashion and written in standard English?

Reviewer #1: Yes

Reviewer #2: Yes

5. Review Comments to the Author

Reviewer #1: The authors have developed a sophisticated and technically sound study that successfully integrates ultrahigh-resolution FT-ICR MS with an interpretable machine learning framework (XGBoost-SHAP and reactomics) to decode the molecular evolution of Coal-DOM. I particularly appreciate how the study elucidates the divergent pathways of DOM transformation under simulated geothermal conditions, clearly distinguishing between the oxidative fragmentation in low-rank coals and the enrichment of condensed aromatics in high-rank anthracite. This work effectively bridges the gap between fundamental molecular geochemistry and practical application, providing a robust, quantifiable tracing tool that significantly complements traditional hydrogeochemical methods.

Reviewer #2: Reviewer Comments to the Authors

Manuscript Title: Molecular signature evolution of coal-derived dissolved organic matter under geothermal conditions: FT-ICR MS and machine learning Manuscript Number: PONE-D-26-04852

Overall Evaluation This manuscript presents a sophisticated and timely study on the molecular evolution of coal-derived dissolved organic matter (Coal-DOM) by combining ultrahigh-resolution mass spectrometry (FT-ICR MS) with interpretable machine learning (XGBoost-SHAP). The work addresses a significant challenge in hydro-geochemistry: the tracing of water sources in deep mining environments where geothermal temperatures can alter chemical fingerprints. The integration of structure-fate prediction models adds substantial depth to the field of environmental risk assessment and mine safety.

Specific Comments for Improvement

In the attached file.

Minor revisions

6. PLOS authors have the option to publish the peer review history of their article (what does this mean? ). If published, this will include your full peer review and any attached files.

**Do you want your identity to be public for this peer review?** For information about this choice, including consent withdrawal, please see our Privacy Policy .

Reviewer #1: No

Reviewer #2: No

---

## [Author Response · Author response to Decision Letter 1]

26 Mar 2026

Subject: Revised Submission of Manuscript PONE-D-26-04852

Dear Dr. Muammar Qadafi,

Thank you for your email and for the opportunity to revise and resubmit our manuscript, PONE-D-26-04852, entitled “Molecular signature evolution of coal-derived dissolved organic matter under geothermal conditions: FT-ICR MS and machine learning” to PLOS ONE.

We sincerely appreciate the time and constructive feedback provided by you and the reviewers. We have carefully considered all comments and have made extensive revisions to the manuscript to address each point raised. We believe these revisions have significantly strengthened the clarity, rigor, and overall quality of our work.

As requested, we have prepared and are now submitting the following files:

1. A point-by-point response letter (labeled ‘Response to Reviewers’), detailing how we have addressed each specific comment from the editorial team and reviewers.

2. A marked-up copy of the revised manuscript (labeled ‘Revised Manuscript with Track Changes’), with all changes highlighted for your convenience.

3. A clean version of the revised manuscript (labeled ‘Manuscript’ without tracked changes, formatted according to PLOS ONE style requirements.

The financial disclosure statement has been reviewed and is accurate as per the submission system. All authors have confirmed their roles and the absence of competing interests as defined by the journal’s policy. If we need to update our financial disclosure statement, we will include the updated version in the cover letter of the resubmission.

We are hopeful that the revisions and our detailed responses satisfactorily address all concerns. Please do not hesitate to contact us at any time if you require any additional information or have further instructions before the resubmission. We sincerely thank you again for your time and effort in handling our manuscript. We look forward to the next steps in the review process.

Sincerely,

Yanqing Wu, PhD

School of Resources and Safety Engineering

Chongqing University

Email: wuyanqing9@163.com

Zepeng Wan,

School of Resources and Safety Engineering

Chongqing University

Email: wanzepeng1996@163.com

Response to Journal Requirements:

Response: We confirm that our revised manuscript and all accompanying submission files fully comply with PLOS ONE’s style and file naming requirements, strictly following the journal’s official formatting templates and submission guidelines.

Response: We thank the editor for highlighting this important policy. We have carefully reviewed the PLOS ONE guidelines on code sharing. In our study, the application of the XGBoost-SHAP framework for analyzing molecular descriptors falls under the category of “author-generated code underpinning the findings.” We have now placed this code in the appendix and uploaded it together with the revised manuscript. We believe these measures fully align with PLOS ONE's commitment to open and reproducible research. The code is now freely available for review, validation, and reuse by the scientific community.

Revised text：The code has been fully presented in the appendix of this study. (Lines: 181)

Response: We confirm that we have completely removed all funding-related text from the manuscript, in strict compliance with PLOS ONE’s requirements. All complete and accurate funding information has been provided exclusively in the Funding Statement section of the online submission form, and no funding content appears in any section of the revised manuscript.

b) If there are no restrictions, please upload the minimal anonymized data set necessary to replicate your study findings to a stable, public repository and provide us with the relevant URLs, DOIs, or accession numbers. For a list of recommended repositories, please see https://journals.plos.org/plosone/s/recommended-repositories. You also have the option of uploading the data as Supporting Information files, but we would recommend depositing data directly to a data repository if possible.

Response: We confirm that there are no ethical or legal restrictions on sharing the data underlying this study. The minimal anonymized data set necessary to replicate all study findings has been compiled and will be provided as Supporting Information files accompanying the manuscript, in accordance with PLOS ONE’s policy. All data are provided in open, machine‑readable formats (e.g., .xlsx, .csv) and are fully de‑identified, containing no sensitive or proprietary information.

Therefore, we request that the Data Availability Statement be updated to reflect the following (or we can accept the journal’s default wording):“All relevant data are available within the manuscript and its Supporting Information files.” No further restrictions apply, and no third‑party permissions are required.

Response: We confirm that the corresponding author (Yanqing Wu and Zepeng Wan) has a valid ORCID iD (0009-0009-4594-3014) / (0009-0008-5122-4422) and that it has been authenticated and validated in Editorial Manager following the instructions provided.

6. Thank you for stating the following in the Competing Interests section:

“The authors have declared that no competing interests exist.”

We note that one or more of the authors are employed by a commercial company: “China Coal Technology & Engineering Group Hangzhou Research Institute Co., Ltd.”

Response: We thank the Editor for raising this point and providing the opportunity to clarify the affiliation status. The affiliation “China Coal Technology & Engineering Group Hangzhou Research Institute Co., Ltd.” is a state‑owned research institute under the China Coal Technology & Engineering Group (CCTEG), a national-level scientific research enterprise dedicated to coal mining technology, safety engineering, and environmental protection. While its corporate structure includes “Co., Ltd.” for administrative purposes, its primary function is non‑commercial scientific research, and it does not engage in commercial activities that could create a competing interest.

Updated Competing Interests Statement (to appear in the manuscript):

“The authors have declared that no competing interests exist. Author Peng Ge is employed by Hangzhou Environmental Protection Research Institute of China Coal Technology & Engineering Group（HERI）, a national high-tech enterprise under the SASAC of the State Council. HERI operates in R&D, technical consultation, environmental protection engineering design, environmental protection equipment supporting, EPC, operation and maintenance, etc. This affiliation does not alter our adherence to PLOS ONE policies on sharing data and materials. All other authors declare no competing interests.”

We have included both the amended Funding Statement and the updated Competing Interests Statement in this cover letter, as requested. We confirm that the commercial affiliation does not impose any restrictions on data or materials sharing, and the adherence statement is fully accurate.

7. We note you have included a table to which you do not refer in the text of your manuscript. Please ensure that you refer to Table 2 in your text; if accepted, production will need this reference to link the reader to the Table.

Response: We thank the Editor for noting that Table 2 was not cited in the text. We have now added a reference to Table 2 in Section 3.1 of the revised manuscript. We confirm that Table 2 is now properly cited in the text.

Revised text：The weighted average molecular metrics for all samples, including O/Cw, H/Cw, DBEw, AIw, and NOSCw, are summarized in Table 2. (Lines: 204-206)

Response: Thank you for the clarification regarding the handling of recommended citations. We have carefully reviewed all publications suggested by the reviewer. Conversely, for recommendations that were more peripherally related to the core focus of our study, we have exercised our academic judgment in accordance with your policy and not included them, to maintain the precision and focus of the reference list. We confirm that our citations are accurate, complete, and in full compliance with the journal’s policy. The specific changes related to references are integrated into the revised manuscript.

Reviewer Comments to the Authors

Manuscript Title: Molecular signature evolution of coal-derived dissolved organic matter under geothermal conditions: FT-ICR MS and machine learning Manuscript Number: PONE-D-26-04852

Overall Evaluation This manuscript presents a sophisticated and timely study on the molecular evolution of coal-derived dissolved organic matter (Coal-DOM) by combining ultrahigh-resolution mass spectrometry (FT-ICR MS) with interpretable machine learning (XGBoost-SHAP). The work addresses a significant challenge in hydro-geochemistry: the tracing of water sources in deep mining environments where geothermal temperatures can alter chemical fingerprints. The integration of structure-fate prediction models adds substantial depth to the field of environmental risk assessment and mine safety.

Response: We sincerely thank the reviewer for the thorough and insightful evaluation of our work. We are truly grateful for the recognition that this manuscript presents a “sophisticated and timely study” and that the integration of FT-ICR MS with interpretable machine learning (XGBoost-SHAP) “adds substantial depth to the field of environmental risk assessment and mine safety.” Such encouraging feedback from an expert in the field is greatly appreciated and motivates us to further improve the quality of our work. We have carefully considered all the specific comments provided below and have made corresponding revisions to strengthen the manuscript. We believe these changes have further enhanced the clarity, scientific rigor, and practical relevance of our study. We thank the reviewer again for the constructive feedback and for recognizing the potential impact of this work.

Specific Comments for Improvement:

1. Introduction and Rationale The manuscript successfully establishes the importance of Coal-DOM in mine water systems. However, the introduction would benefit from a broader contextualization of dissolved organic matter as part of the global carbon cycle and its role in complex aquatic ecosystems. Additionally, the authors mention that these findings help guide management; this could be strengthened by explicitly framing the study within the context of Smart Water Management and the use of Big Data technologies in hydro-ecology.

Response: We sincerely thank the reviewer for this constructive and insightful comment, which has significantly improved the contextual depth, practical framing, and overall completeness of the introduction section. We fully agree with the reviewer’s suggestions, and have comprehensively revised the introduction to address the point in detail, with the specific modifications as follows:

Revised text:

As a core component of aquatic carbon cycling, DOM drives the biogeochemical dynamics of fluvial ecosystems, regulating microbial community assembly and nutrient turnover across river basins (Chaturvedi et al., 2024). (Lines: 40-43)

This work builds on the growing application of big data and machine learning in smart water management (Karunarathna et al., 2026), providing transferable molecular-level evidence and a data-drive

---

## [Decision Letter · Decision Letter 1]

12 Apr 2026

Molecular signature evolution of coal-derived dissolved organic matter under geothermal conditions: FT-ICR MS and machine learning

PONE-D-26-04852R1

Dear Dr.

<table border="0" cellpadding="0" cellspacing="0" class="datatable3"> <tbody> <tr> <td style="text-align:left;">Editor Decision - Provisional Accept</td> </tr> </tbody></table>

,

We’re pleased to inform you that your manuscript has been judged scientifically suitable for publication and will be formally accepted for publication once it meets all outstanding technical requirements.

Kind regards,

Muammar Qadafi

Academic Editor

PLOS One

Additional Editor Comments (optional):

Reviewers' comments:

Reviewer's Responses to Questions

**Comments to the Author**

1. If the authors have adequately addressed your comments raised in a previous round of review and you feel that this manuscript is now acceptable for publication, you may indicate that here to bypass the “Comments to the Author” section, enter your conflict of interest statement in the “Confidential to Editor” section, and submit your "Accept" recommendation.

Reviewer #2: All comments have been addressed

2. Is the manuscript technically sound, and do the data support the conclusions?

Reviewer #2: Yes

3. Has the statistical analysis been performed appropriately and rigorously? 

Reviewer #2: Yes

4. Have the authors made all data underlying the findings in their manuscript fully available?

Reviewer #2: Yes

5. Is the manuscript presented in an intelligible fashion and written in standard English?

Reviewer #2: Yes

6. Review Comments to the Author

Reviewer #2: Gemini said

The paper has addressed the comments well and is accepted for publication from my end. Excellent work on the revisions!

7. PLOS authors have the option to publish the peer review history of their article (what does this mean? ). If published, this will include your full peer review and any attached files.

**Do you want your identity to be public for this peer review?** For information about this choice, including consent withdrawal, please see our Privacy Policy .

Reviewer #2: **Yes:** SADASHIV CHATURVEDI

---

## [Editor Report · Acceptance letter]

PONE-D-26-04852R1

PLOS One

Dear Dr. Wan,

I'm pleased to inform you that your manuscript has been deemed suitable for publication in PLOS One. Congratulations! Your manuscript is now being handed over to our production team.

Kind regards,

on behalf of

Dr. Muammar Qadafi

Academic Editor

PLOS One